# The Role of Omega-3 Polyunsaturated Fatty Acids and Their Lipid Mediators on Skeletal Muscle Regeneration: A Narrative Review

**DOI:** 10.3390/nu15040871

**Published:** 2023-02-08

**Authors:** Sebastian Jannas-Vela, Alejandra Espinosa, Alejandro A. Candia, Marcelo Flores-Opazo, Luis Peñailillo, Rodrigo Valenzuela

**Affiliations:** 1Instituto de Ciencias de la Salud, Universidad de O’Higgins, Rancagua 2820000, Chile; 2Escuela de Medicina, Campus San Felipe, Universidad de Valparaíso, San Felipe 2170000, Chile; 3Exercise and Rehabilitation Sciences Institute, School of Physical Therapy, Faculty of Rehabilitation Sciences, Universidad Andres Bello, Las Condes, Santiago 7591538, Chile; 4Department of Nutrition, Faculty of Medicine, University of Chile, Santiago 8380000, Chile

**Keywords:** omega-3, skeletal muscle, oxylipins, endocannabinoids, regeneration

## Abstract

Skeletal muscle is the largest tissue in the human body, comprising approximately 40% of body mass. After damage or injury, a healthy skeletal muscle is often fully regenerated; however, with aging and chronic diseases, the regeneration process is usually incomplete, resulting in the formation of fibrotic tissue, infiltration of intermuscular adipose tissue, and loss of muscle mass and strength, leading to a reduction in functional performance and quality of life. Accumulating evidence has shown that omega-3 (n-3) polyunsaturated fatty acids (PUFAs) and their lipid mediators (i.e., oxylipins and endocannabinoids) have the potential to enhance muscle regeneration by positively modulating the local and systemic inflammatory response to muscle injury. This review explores the process of muscle regeneration and how it is affected by acute and chronic inflammatory conditions, focusing on the potential role of n-3 PUFAs and their derivatives as positive modulators of skeletal muscle healing and regeneration.

## 1. Introduction

Skeletal muscle is the largest tissue in the human body, comprising approximately 40% of body mass, and has an essential role in energy metabolism, locomotion, and stability [1]. These actions can be compromised by muscle damage or injury resulting from a variety of events, including lacerations, contusions, strains, or exercise [2]. Following damage, a healthy skeletal muscle is often fully regenerated; however, with aging and chronic diseases, such as obesity, type 2 diabetes, rheumatoid arthritis, chronic obstructive pulmonary disease, and muscular dystrophies, the regeneration process may be incomplete, resulting in the formation of fibrotic tissue, infiltration of intermuscular ectopic adipose tissue, and loss of muscle mass and strength [3,4,5,6], which might lead to a reduction in functional performance and quality of life.

Satellite cells (SCs) are considered the major players in the regeneration process after muscle injury [7]. They are located between the sarcolemma and the basal lamina, representing approximately 2–7% of skeletal muscle cells [3,8,9]. In homeostatic conditions, these cells reside in their niche in a quiescent state, whereas upon muscle damage, SCs become activated and proliferate to form myogenic precursor cells (i.e., myoblasts). An efficient regenerative capacity is supported by the maintenance of a promyogenic muscle niche by additional cells in the local milieu, including mesenchymal fibroadipogenic progenitors (FAPs), immune cells (i.e., eosinophils, macrophages, Treg), and endothelial cells [10,11]. In the presence of a favorable promyogenic cellular microenvironment, most of the SCs differentiate and form new myocytes that fuse with damaged myofibers, leading to muscle tissue repair, while a fraction of the SC population, generated by asymmetric division, self-renew and return to quiescence. In parallel, a tightly regulated and time-dependent recruitment of immune cells occurs, releasing inflammatory factors (e.g., TNF- α, IL-6), which also have the capacity to promote SC activation and proliferation. Later in the regeneration process, immune cells together with FAP cells contribute to the removal of cell debris and necrotic tissue [12,13,14]. This response is followed by the clearance of proinflammatory cytokines and the recruitment of anti-inflammatory immune cells, promoting SC differentiation, tissue repair, and the return of tissue homeostasis [15,16,17].

In chronic inflammatory conditions, this process is usually dysregulated and impaired, preventing the healing of damaged tissue, leading to loss of muscle function and decreased quality of life [18,19]. In consequence, it is of major importance to develop strategies that can mitigate and adequately regulate the inflammatory response after muscle injury. Accumulating evidence has shown that the omega-3 (n-3) polyunsaturated fatty acids (PUFAs) and their lipid mediators (i.e., oxylipins and endocannabinoids) have the potential to enhance muscle regeneration by positively modulating the local and systemic inflammatory response to muscle injury [20,21,22]. This review will explore the process of muscle regeneration and how it is affected by acute and chronic inflammatory conditions, focusing on the potential role of n-3 PUFAs and their derivatives as positive modulators of skeletal muscle healing and regeneration.

## 2. An Inflammatory Process Initiates Skeletal Muscle Repair

The inflammatory response triggered by an acute muscle injury is required to command SCs proliferation and myogenesis, and coordinates scavenger activity and phagocytosis to properly eradicate cellular debris and progress to tissue regeneration (Figure 1). The initiation, development, and resolution of the inflammatory response involves interactions between circulating and resident immune cells and SCs within the muscle tissue [11], of which the most abundant are mast cells and macrophages [13]. These resident cells act as the primary responders to injury by secreting proinflammatory molecules, including tumor necrosis factor alpha (TNF-α), and macrophage inflammatory protein-2 (MIP-2) [23]. This initial burst promotes vasodilation, vascular permeability, and rapid recruitment of neutrophils, which ingest and remove cellular debris from the damaged tissue and release a myriad of chemokines and cytokines that attract the recruitment of monocytes [24], and activation of FAPs via the interleukin 4 signaling pathway [14]. The infiltrated monocytes divide into two main categories: (1) Ly6C+ monocytes, which peak during the first few days of injury (days 1–3); and (2) the Ly6C− monocytes, which are recruited later (days 3–7) for the optimum regeneration of muscle tissue. The Ly6C+ and Ly6C− monocytes rapidly differentiate into the proinflammatory M1 and anti-inflammatory M2 macrophages, respectively. The M1 macrophage phenotype expresses proinflammatory cytokines (IL-1β, IL-6, IL-8, and TNF-α), responsible for the activation and proliferation of SCs, while the M2 macrophage phenotype is responsible for the release of anti-inflammatory molecules such as IL-10 and transforming growth factor-beta (TGF-β), promoting the differentiation and fusion of myotubes into damaged myofibers, leading to proper regeneration [11]. On the other hand, the activated resident FAPs will proliferate within a narrow time window, generally no longer than 5 days postinjury, and secrete a plethora of promyogenic cytokines and growth factors, including Follistatin, IL-6, WNT1 inducible signaling pathway protein 1 (WISP1), and IGF-1 [25]. Later, FAP numbers rapidly return to basal levels by the induction of apoptosis, a mechanism mediated by the release of TNF- α by M1 macrophages [26].

The inflammatory response to damage also leads to the production of reactive oxygen species (ROS) and reactive nitrogen species (RNS) (collectively known as RONS), from circulating and resident immune cells. The RONS are molecules with one or more unpaired electrons in atomic or molecular orbitals derived from oxygen or nitrogen [27]. The main RONS formed in cells include superoxide (O2-), and nitric oxide (NO). Both O2- and NO can rapidly combine to produce peroxynitrite, and its reaction is three times faster than the dismutation of superoxide to hydrogen peroxide via superoxide dismutase (SOD) [28]. In healthy muscle, RONS activate signaling pathways essential for proper muscle regeneration [27]; however, when RONS are exacerbated (i.e., chronic inflammatory conditions), it can lead to impaired regeneration through the inhibition of myogenesis, cell death, and loss of muscle function [29]. Collectively, these processes highlight the importance of an efficient and coordinated inflammatory response after muscle injury, as slight shifts in the inflammatory and oxidative stress responses could decrease muscle regenerative capacity as observed in obese and aged populations.

A crucial yet overlooked metabolic process regulating the inflammatory response after muscle damage is the synthesis of bioactive lipid metabolites derived from n-3 and n-6 PUFAs, the oxylipins and endocannabinoids. These lipid mediators are suggested to regulate the muscle regeneration process via autocrine and paracrine inflammatory signaling of immune cells.

## 3. Role of PUFAs on Inflammation Resolution

Fatty acids (FAs) have diverse functions of physiological importance as they are major components of cellular membranes, precursors for the synthesis of bioactive lipids, and major sources of energy [30]. Saturated FAs contain no double bonds in their hydrocarbon chain, have a rigid structure, and are the most abundant FAs in the human diet, with palmitic and stearic acid as the most popular. Conversely, unsaturated FAs have one or more double bonds in their hydrocarbon chain; those with one double bond are classified as monounsaturated fatty acids (MUFAs). The most common MUFAs are omega-7 (n-7) and omega-9 (n-9), with palmitoleic acid and oleic acid being the most consumed in the diet, respectively. Both MUFAs have been reported to reduce the risk of heart disease and inflammation [31,32].

FAs with two or more double bonds are classified as polyunsaturated fatty acids (PUFAs). PUFAs are mainly composed by n-3 and n-6 fatty acids. N-3 PUFAs have their first double bond on the third hydrocarbon chain when counted from the methyl terminus, whereas the n-6 PUFAs have their first double bond on the sixth hydrocarbon chain. The shortest members of each family are α-linolenic acid (C18:3n-3, ALA) and linoleic acid (C18:2n-6, LA), which are 18 hydrocarbons in acyl-chain length [33,34,35]. Through a series of reactions catalyzed by the same enzymes in the liver, ALA and LA can be further metabolized by desaturation and elongation to eicosapentaenoic acid (C20:5n-3, EPA) and docosahexaenoic acid (C22:6n-3, DHA), and to arachidonic acid (C20:4n-6, AA), respectively [35,36]. As n-6 PUFAs are consumed in the diet 15 times more than n-3 PUFAs, this process results in a higher synthesis of AA at the expense of EPA and DHA [30]. In consequence, the presence of n-3 PUFAs in cell membranes and tissues is typically low and depends on the amount of EPA and DHA ingested in the diet. Thus, when consumed in high quantities, EPA and DHA are incorporated into cell membrane phospholipids [37], resulting in a decreased synthesis of n-6 PUFAs derivatives and an increased synthesis of n-3 PUFA-lipid mediators [38]. As the n-6 PUFAs are associated with proinflammatory actions [39] and the n-3 lipid derivatives are known for their anti-inflammatory and pro-resolving actions [40,41,42], their concentrations in cell membranes could determine the regeneration process after muscle injury.

There is emerging evidence that bioactive lipids derived from n-3 and n-6 PUFAs play a key role in the initiation and resolution of the inflammatory response [12,43]. After a muscle injury, n-3 and n-6 PUFAs are rapidly released from immune cell membrane phospholipids via phospholipase enzymes and metabolized via enzymatic reactions to the lipid mediators oxylipins and endocannabinoids [44]. During the early stages of injury, the classical n-6 PUFA-derived lipid mediators are synthesized and released, promoting acute inflammation by regulating local blood flow, vascular permeability, cytokine production, and leukocyte chemotaxis [12]. Later, a shift in the profile of these mediators results in the generation of oxylipins and endocannabinoids mainly derived from n-3 PUFAs, whose functions are to actively resolve and terminate inflammation, leading to tissue regeneration and return to homeostasis [45,46].

The pathways involved in the formation of oxylipins and endocannabinoids are a complex network of time-specific enzymatic reactions, and to date have not been fully elucidated (Figure 2). These metabolites act as intercellular messengers and mediators of the muscle regeneration process by regulating the inflammatory response to injury [12]. The oxylipins represent the most common and widest family of bioactive lipids synthesized from the long-chain n-6 and n-3 precursors, including LA, AA, ALA, EPA, and DHA [45]. These PUFAs are released upon endogenous and exogenous stimuli from membrane phospholipids via phospholipase A2, although they can be released by phospholipase C, and metabolized into their bioactive products via cyclooxygenase (COX), lipooxygenase (LOX), and cytochrome P450 (CYP450) enzymes [15]. Once formed, oxylipins can mediate their biological effects via interactions with G protein-coupled receptors or intracellular effectors, including peroxisome proliferator-activated receptor gamma (PPARγ) [15]. The oxylipins derived from AA (n-6) are the most common and include the two-series prostaglandins (PGs), thromboxanes (TXs), and the four-series leukotrienes (LTs); meanwhile, EPA (n-3) is the precursor of the three-series PGs and TXs and five-series LTs. The PGs, TXs, and LTs are all proinflammatory in nature; however, those that are EPA-derived are less potent compared with those synthesized from AA [44]. Oxylipins with anti-inflammatory and proresolving activities are mostly derived from EPA and DHA, including the resolvins (Rv) from EPA (E series) and DHA (D series) and the DHA-derived maresins (Ma) and protectins (Pr), collectively termed “specialized pro-resolving mediators” (SPMs). AA-derived lipoxins also contribute to this group [43].

The family of endocannabinoids consists of the precursors 2-acylglycerols and ethanolamides, of which the most abundant and best characterized are the n-6-derived 2-arachidonoylglycerol (2-AG) and n-arachidonoylethanolamine (AEA, anandamide), respectively [47]. The 2-AG are agonists with low-to-moderate affinity for cannabinoid receptors 1 and 2 (CB1-2), while AEA is a partial agonist with a higher affinity to bind to CB1 relative to CB2. Both CB1 and CB2 are expressed in the central nervous system (CNS) and peripheral tissues [48]. CB1 are predominantly expressed in the CNS and are responsible for mediating neurobehavioral activities such as the regulation of appetite and executive functions [49]. Meanwhile, CB2 are mainly expressed in peripheral tissues, including cells from the immune system, and therefore are in part responsible for mediating the inflammatory response to injury or pathogens [50]. Notably, exogenous anandamide during lactation increases body fat content and CB1 receptor levels in adipose tissue [51].

Recent evidence has shown that the n-3 PUFAs can also be converted into 2-acylglycerols and ethanolamides, including eicosapentanoylglycerol (EPG), eicosapentaenoyl ethanolamide (EPEA), docosahexaenoyl ethanolamide (DHEA), and docosahexanoyl-glycerol (DHG) [52]. Moreover, they can be further metabolized by COX, LOX, and CYP450 enzymes into bioactive endocannabinoid epoxides [53,54], agonists with greater affinity for CB receptors than their former metabolites [46]. As the endocannabinoids derived from n-3 and n-6 PUFAs have been shown to have anti-inflammatory activity, they could play an important role in the muscle regeneration process; however, evidence to date is scarce.

Overall, oxylipins and endocannabinoids are synthesized from membrane-bound n-3 and n-6 PUFAs of immune cells and peripheral tissues. These metabolites share some actions, including the immune cells’ inflammatory response to stress or injury, and thus play an important role in the muscle regeneration process [20,21,46]. Indeed, in vitro and rodent studies have recently shown that n-3 PUFAs and their derived metabolites positively modulate the inflammatory response to muscle injury [21,55,56,57,58,59]. Furthermore, daily ingestion of >1 g/d EPA and DHA increases n-3-derived oxylipin [38] and endocannabinoid [60,61] levels in humans. Interestingly, when administered in periods longer than eight weeks, EPA and DHA can increase muscle protein synthesis [62] and significantly reduce symptoms after eccentric exercise-induced damage [63], which may indicate an enhanced muscle regeneration. In turn, it has been described that oxylipin levels depend on the skeletal muscle type; for example, in the soleus, a muscle with predominant composition (~80%) of slow-twitch fibers [64], there is more oxylipin content than in the gastrocnemius muscle, which contains a mixed-fiber composition [65]. The latter suggests that the accumulation of oxylipins could be regulated according to muscle fiber type. This fact could be relevant in the phenotypic muscle fiber shift observed during ageing, that features a distinctive fast-to-slow fiber type transition and subsequent muscle weakness [66]. On this basis, we speculate that the improvements in muscle regeneration observed after n-3 PUFA supplementation could be in part mediated by increased production of n-3 PUFA-derived oxylipins and endocannabinoids, which may reflect a redistribution of muscle fiber composition.

## 4. Potential Beneficial Effects of n3-PUFA-Derived Metabolites in Muscle Regeneration: Evidence from In Vitro Muscular Cell Lines, Adult Skeletal Fibers, and Animal Models

Following myofiber damage, a rise in the production of proinflammatory n-3 and n-6 PUFA-derived oxylipins and endocannabinoids is necessary for the proper regulation of muscle regeneration [21]. However, in chronic proinflammatory conditions, commonly observed with aging and metabolic diseases, the muscle regeneration process is impaired and linked to fibrosis and infiltration of lipids [19]. Moreover, in aging, there is a deficiency of intramuscular pro-resolving lipid mediator biosynthesis, and RvD1 treatment does not rescue age-related defects in myofiber regeneration [67]. These alterations may be associated with an overproduction of n-6 PUFA-derived metabolites and a decreased synthesis of the bioactive pro-resolving anti-inflammatory lipids oxylipins and endocannabinoids derived from the n-3 PUFAs. In this context, a rise in the concentrations of n-3 PUFA-derived metabolites, through administration of EPA, DHA, and/or n-3 PUFA-derived metabolites, could be a potential strategy to enhance muscle regeneration [20,21,46].

The incorporation of n3-PUFA into cell membranes affects the skeletal muscle proliferation and differentiation processes via changes in membrane fluidity, membrane microdomains involved in cellular signaling, and via the regulation of inflammation through the production of n-3 PUFA-derived metabolites [37,68]. Interestingly, to date, there is little evidence regarding the beneficial role of n3-PUFAs and their derivatives in the myogenic process, mostly because experiments have been performed in uninjured or inflammation-free conditions [69]. In these conditions, both EPA and DHA downregulate the proliferation and differentiation of C2C12 skeletal muscle cells when compared to a fatty-acid free control condition [55,56,70,71,72]. These effects appear to be dose- and time- dependent with higher doses (>50 µM) and longer incubation times (>48 h), yielding increased inhibition of SC proliferation and differentiation, which has been suggested to be mediated by the accumulation of lipid droplets [70]. To the best of our knowledge, only one study has examined the role of the n-3 PUFA-derived metabolites on myogenesis [21]. Murine C2C12 cells incubated with a supraphysiological dose of RvD1 (1µM) resulted in increased myotube diameter [21]. Similarly, mature C2C12 myotubes incubated with very high concentrations of EPA (400–600 µM) and DHA (300–700 µM) decreased protein degradation through partial inhibition of the proinflammatory NF-κB pathway [73]. Thus, it appears that very high doses of n-3 PUFAs and/or their derived metabolites promote myofiber hypertrophy by decreasing inflammation and protein degradation.

In vitro treatment with proinflammatory molecules (e.g., palmitate, TNF-α, lipopolysaccharide [LPS]) mimics some of the metabolic abnormalities associated with chronic inflammation, including decreased protein synthesis, increased protein degradation, and muscle atrophy [57,74,75,76]. Under these conditions, the effects of n-3 PUFA and their derivatives on myogenesis are promising. First, DHA treatment protects against muscle palmitate-induced myofiber atrophy [76,77,78]. Similarly, the addition of 25 µM of EPA or DHA suppresses the decline in myotube diameter and myofibrillar protein content induced by LPS [79], and cotreatment with 700 µM PA and 50 µM of EPA or DHA blunts the expression of IL-6 and TNF-α induced by PA alone in C2C12 myotubes [80]. Secondly, exposure to EPA-derived oxylipin RvE1 decreased LPS-induced IL-6 and MCP-1 expression in C2C12 myotubes [58]. Moreover, RvE1 attenuated secreted IL-6 protein levels and prevented LPS-induced myotube atrophy. Likewise, the DHA-derived oxylipin RvD1 protected against TNF-α-mediated myotube atrophy [21]. Mechanistically, these effects appear to be mediated in part by the restoration of the Akt/mTOR/FoxO3 pathway involved in the muscle differentiation process [76,77], as well as the inhibition of the proinflammatory transcription factors activation protein-1 and NF-κB [80,81,82]. Moreover, in dystrophic myoblasts derived from dystrophin-deficient mdx mice exposed to proinflammatory macrophage-conditioned medium, treatment with RvD2 enhanced myoblast differentiation compared to control and prednisone-treated groups [83]. This effect was associated with a 2-fold increase in myogenin-expressing differentiated myoblasts along with a concomitant decrease in the proportion of undifferentiated Pax7+ cells. In addition, RvD2 increased the fusion index and the expression of myosin heavy chain (MyHC), a marker of terminal differentiation, while knockdown of the cannabinoid receptor Gpr18 blocked RvD2 promyogenic effects via downregulation of the Akt-mTOR pathway [83].

Altogether, these results show the promising effects of n-3 PUFAs and their derived metabolites, specifically the oxylipins, to sustain myogenesis during inflammatory conditions, at least in part via activation of the Akt-mTOR pathway and inhibition of proinflammatory signals. Whether n-3 derived endocannabinoids share similar actions in muscle cells remains to be elucidated; however, there is a growing body of evidence showing that these metabolites have greater anti-inflammatory properties compared with EPA, DHA, or the n-6-derived endocannabinoids. Future studies should determine whether n-3 and n-6-derived endocannabinoids positively regulate the myogenic process.

The positive effects of n-3 PUFAs and their derived metabolites on myogenesis and regeneration in muscle cells have also been observed in rodent muscle tissue. For example, Machado et al. [59] observed that 14-day-old mdx mice—a model of Duchenne muscle dystrophy—treated with 300 mg/kg of EPA for 16 days resulted in a decrease in plasma creatine kinase levels and TNF-α muscle protein content concomitantly with a decline in myonecrotic fibers. Further studies from this group corroborated these findings and provided new evidence showing that EPA and/or DHA in mdx mice attenuated the loss of muscle function [84], and increased muscle regenerative capacity by augmenting the levels of MyoD [85]. In parallel, this treatment increased M1-to-M2 macrophage phenotype transition [86,87], decreased inflammation via changes in serum levels of proinflammatory (IFN-γ) and anti-inflammatory (IL-10) cytokines [86], and diminished muscle oxidative stress through downregulation of inducible nitric oxide synthase protein [86] and 4-HNE-protein adducts [84]. Not surprisingly, similar findings have also been observed in other rodent models of muscle damage such as high-fat diet [88] and cardiotoxin-induced gastrocnemius muscle injury [89]. Nevertheless, to our knowledge, no study has identified whether the positive changes in myogenesis observed after n-3 PUFA administration are mediated by their derived metabolites, the oxylipins and endocannabinoids; however, recent evidence has shown a promising role of the oxylipin derived from n-3 PUFA, the RVs, as a potential molecule to enhance muscle regeneration.

Mdx mice treated with 5 µg/kg/d of RvD2 or prednisone, the gold-standard treatment of Duchenne muscle dystrophy, showed reduced neutrophil accumulation and levels of the proinflammatory M1-macrophages (40–50%), and concomitantly increased presence of proregenerative M2- macrophages [83]. Surprisingly, while prednisone administration did not affect the pool of myogenic cells, RvD2 administration induced a ~2-fold increase in the total number of myogenic cells and 2–3-fold increase in the number of Myog+ differentiated myoblasts. Furthermore, assessment of global physical function using the hang test revealed that RvD2 treated mice showed greater global physical function on days 7 and 21 compared with prednisone-treated mdx mice [83]. In a mouse model of barium chloride (BaCl_2_)-induced muscle injury, an intraperitoneal injection of RvD1 resulted in diminished accumulation of CD68+ macrophages; reduced mRNA expression of the proinflammatory molecules Il-6, IL-1β MCP-1, and TNF-α; expedited clearance of polymorphonuclear cells; and enhanced macrophage phagocytosis [21]. These changes translated in enhanced myofiber regeneration and improved recovery (+15%) of muscle strength. Similarly, cardiotoxin-induced muscle (i.e., tibialis anterior, TA) injury, intramuscular administration of 200 pg RvD2 increased M2 and decreased M1 macrophages after 2 and 3 days of muscle injury [20]. As a result, RvD2 administration improved muscle (i.e., TA) force recovery by 50% and muscle mass by 17% after 8 and 14 days of cardiotoxin injury. Finally, in aged mice, daily intraperitoneal injection of RvD1 after intramuscular injection of BaCl_2_ blunted inflammatory cytokine expression and accumulation of fibrotic tissue in TA muscle [67]. This strategy improved muscle-specific force recovery. However, myofiber regeneration was not enhanced when assessed by centrally located nuclei and expression of embryonic myosin heavy chain.

Collectively, these results suggest that in different rodent models of muscle damage, n-3 PUFAs and the D-series resolvins may enhance skeletal muscle regeneration via decreased inflammation and reduced oxidative stress. Whether other oxylipins and endocannabinoids derived from the n-3 PUFAs, including the RvE series, MaRs, PrTs, EPG, EPEA, and DHEA, play a role in the muscle regeneration process remains to be elucidated, as well as whether the beneficial effects of n-3 PUFAs on myogenesis are mediated by an increased production of oxylipins and endocannabinoids.

## 5. Clinical Interventions Supporting the Consumption of n-3 PUFAs to Aid Muscle Recovery

A common approach to investigating the muscle regeneration capacity is by inducing damage through repeated efforts of maximal to near-maximal eccentric lengthening contractions—a force applied to the muscle that exceeds the torque produced by the muscle itself [90]. The high mechanical stress induced by these contractions leads to focal microlesions of the sarcomeres as well as the extracellular matrix and connective tissue of the muscle fibers [91], which manifests itself by a range of clinical symptoms such as delayed-onset muscle soreness (DOMS), muscle stiffness, swelling, decreased proprioceptive function, and loss in maximal force-generating capacity [92]. Moreover, eccentric exercise-induced muscle damage (EIMD) leads to systemic and local inflammatory responses that initially were considered detrimental, albeit similar to the response to pathogens or local injuries. It is now well established that the inflammatory stages are crucial for optimal recovery, as they ensure the removal of tissue debris and promote muscle regeneration via the regulation of different immune cells and activation of SCs. In this context, n-3 PUFAs and their lipid mediators may play a significant role in this process.

The effect of n-3 PUFA supplementation after eccentric damaging exercise has been assessed in some studies. Interestingly, the majority have been performed in healthy young to middle-aged males and have used a wide range of supplementation doses (<1 g up to 6 g/day) and supplementation times (from days to 8 weeks). Despite this, most of the studies have shown that n-3 PUFAs induce slightly faster recovery of muscle function and muscle soreness after EIMD [63,93,94,95,96,97,98,99,100,101]. For instance, Kyriakidou et al. [102] showed that 4 weeks of n-3 PUFA supplementation successfully attenuated minor aspects of EIMD, although it did not improve performance. Recent systematic reviews and meta-analyses have corroborated the efficacy of n-3 PUFAs in reducing DOMS and markers of muscle damage [103,104]; however, only a few studies have found significantly lower maximal muscle strength loss or recovery (i.e., maximal voluntary contraction; MVC), indicated as the best indirect marker of muscle damage, probably because of a poor control of the participants’ diet characteristics affecting the n-6/n-3 ratio and the use of a broad range of supplementation doses and times. Therefore, more studies assessing MVC after damage are warranted.

Systemically, EIMD is paralleled by an inflammatory response involving many mediators, such as interleukin −1 receptor antagonist (IL-1ra), interleukin (IL)-6, IL-10, and acute phase proteins [105,106]. Kyriakidou et al. [102] supplemented healthy young individuals with 3.9 g/d of fish oil containing 3 g of n-3 PUFA (2.145 g of EPA and 0.858 g DHA) per day for a period of 4 weeks (*n* = 7) and induced damaging exercise (downhill running; 60 min at 65% V̇O2max with a −10% gradient), and found a reduced increase in IL-6 and a small protective effect of supplementation with n-3 supplementation in muscle function markers. To the best of our knowledge, no study has yet associated the faster recovery rates after n-3 PUFA supplementation with changes in the levels of oxylipins and endocannabinoids. Future research is warranted to determine the effect of longer time supplementations and larger sample sizes.

The positive effects of n-3 PUFAs have also been observed in patients with Duchenne muscular dystrophy (DMD). These patients lack dystrophin, an important structural skeletal muscle protein. This leads to progressive muscle weakness, chronic muscle degeneration, infiltration of fat, and fibrosis, resulting in the loss of muscle mass and aberrant muscle regeneration, decreasing muscle function, and causing premature death. In a placebo-controlled, double-blinded, randomized study, 28 patients with DMD were supplemented with 2.9 g/d of n-3 PUFAs (*n* = 14) or sunflower oil (placebo, *n* = 14) for 6 months. the results showed that there was a tendency to decrease the loss of lean mass in patients supplemented with n-3 PUFAs compared with the placebo group [107]. In another study by the same group and same set of patients, leukocyte levels of IL-1β and IL-6 were decreased after n-3 PUFA supplementation [108]. In support of these findings, in a 24-week supplementation with a multi-ingredient nutraceutical, including high concentrations of DHA (1.2 g/d) and EPA (0.36 g/d), there was a reduced 6 min walk distance and increased isokinetic knee extension in facioscapulohumeral and limb girdle muscle dystrophy patients when compared to a placebo group [109]. Serum CK levels decreased in all treated groups, with a significant difference in DMD subjects. Conversely, an n3-PUFA-rich diet performed worse than a MUFA-rich diet in MDX mice, suggesting that n3 PUFA may exacerbate stress in dystrophic skeletal muscle [110] potentially by increased fluidity of muscle membranes. However, in general, the evidence provided suggests that EPA and DHA slow the progression of muscle loss and decrease muscle damage potentially via enhanced muscle regeneration through the attenuation of proinflammatory mediators (Figure 3). Whether these changes are mediated by n-3 PUFA oxylipins or endocannabinoids remains to be elucidated.

## 6. Conclusions

We have summarized the existing data that support the potential role of n-3 PUFAs and their lipid mediators (i.e., oxylipins and endocannabinoids) on skeletal muscle healing and regeneration. There is a gap in knowledge regarding which n-3 PUFAs (EPA, DHA, or a combination of both) and specific lipid mediators are involved in this process. Future human studies should be focused in establishing the relationship between changes in membrane composition, endocannabinoid and oxylipin levels, and the changes in regeneration capacity after muscle-damaging protocols using a long-term n-3 PUFA supplementation period (>3 months) of at least 1 g/d (EPA and/or DHA) in healthy and/or diseased populations.

## Figures and Tables

**Figure 1 nutrients-15-00871-f001:**
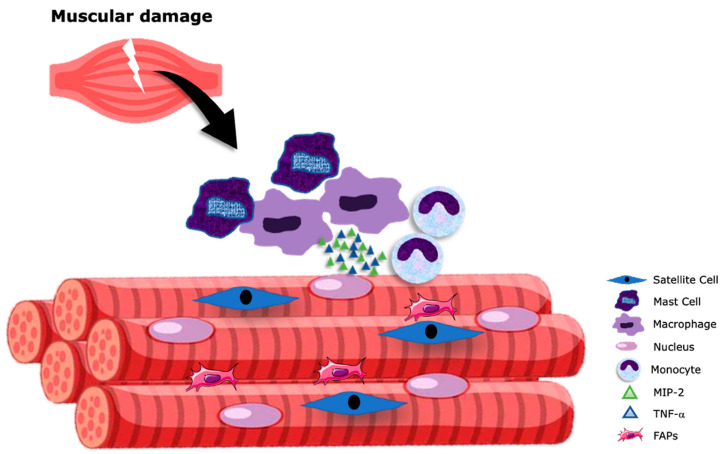
Inflammatory response to muscle damage. After an acute muscular injury, resident cells (mast cells and macrophages) secrete proinflammatory molecules such as tumor necrosis factor alpha (TNF-α) and macrophage inflammatory protein-2 (MIP-2), promoting neutrophil and monocyte recruitment to the injury region. FAPs: fibroadipogenic precursors. The Figure was partly generated using Servier Medical Art, provided by Servier, licensed under a Creative Commons Attribution 3.0 unported license.

**Figure 2 nutrients-15-00871-f002:**
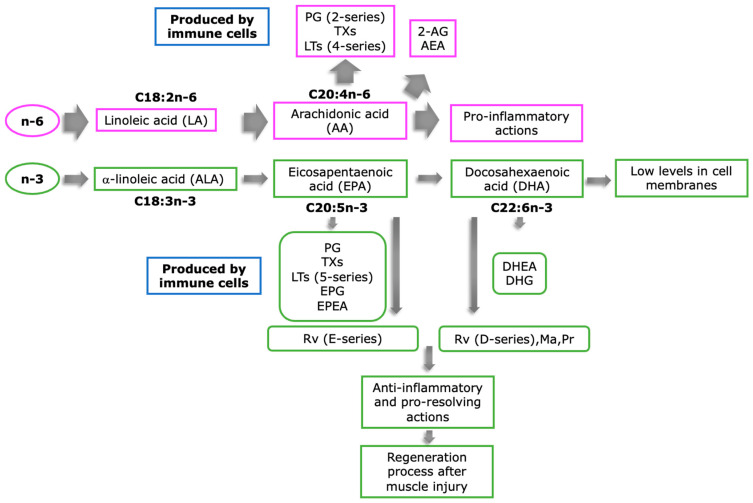
Omega-6 (n-6) and omega-3 (n-3) polyunsaturated fatty acid (PUFA) lipid mediators. The n-6 PUFA, linoleic acid (LA), is elongated and desaturated in the liver to arachidonic acid (AA). AA is processed by immune cells and converted into the two-series prostaglandins (PGs), thromboxanes (Txs), the four-series leukotrienes (LTs), and to the endocannabinoids, 2-acylglycerols (2-AG) and n-arachidonoylethanolamine (AEA). This metabolic pathway is associated with proinflammatory actions. On the other hand, the n-3 PUFA, α-linoleic acid (ALA), is elongated and desaturated in the liver to eicosapentaenoic acid (EPA) and docosahexaenoic acid (DHA). Immune cells convert EPA to the three-series PG, TXs, five-series LTs, to the endocannabinoids, eicosapentanoylglycerol (EPG), eicosapentaenoylethanolamide (EPEA), and to the e-series derived resolvins (Rv). DHA is converted to docosahexaenoyl ethanolamide (DHEA) and docosaheanoyl-glycerol (DHG) and to the D-series Rv, maresins (Ma) and protectins (Pr). The n-3-derived lipid mediators have anti-inflammatory and proresolving actions, which may accelerate the muscle regeneration process.

**Figure 3 nutrients-15-00871-f003:**
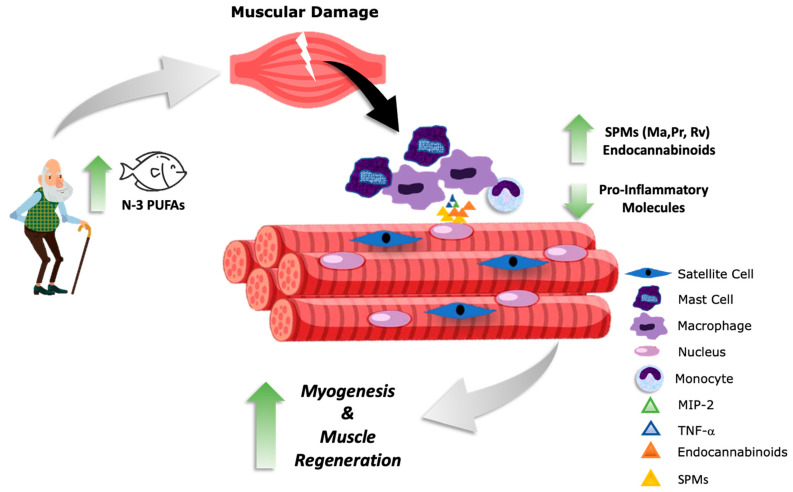
Potential benefits of the omega-3 (n-3) polyunsaturated fatty acid (PUFA) lipid mediators after muscular damage. N-3 PUFA consumption leads to increased synthesis of endocannabinoids and specialized proresolving mediators (SPMs) such as resolvins (Rv), maresins (Ma), and protectins (Pr), leading to a decrease in the production of proinflammatory molecules. These effects may accelerate inflammation resolution, improving myogenesis and muscle regeneration. TNF-α: tumor necrosis factor alpha; MIP-2: macrophage inflammatory protein-2. The Figure was partly generated using Servier Medical Art, provided by Servier, licensed under a Creative Commons Attribution 3.0 unported license.

## Data Availability

No new data were created or analyzed in this study. Data sharing is not applicable to this article.

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
