# Peer review of "The Role of Omega-3 Polyunsaturated Fatty Acids and Their Lipid Mediators on Skeletal Muscle Regeneration: A Narrative Review"

_nutrients, 2023, doi:10.3390/nu15040871_

Round 1

Reviewer 1 Report

This top-level review discusses the advantages of omega-3 fatty acids on muscle health. I have some remarks:

- In the clinical part (paragraph 5), more emphasis should be put on the n-6/n-3 balance (as has been done in paragraph 3), as this explains (partially) the discrepancies between clinical studies.

- While the n-6/n-3 balance is already quite well documented, I would also propose to add a short piece concerning n-7 and n-9 fatty acids.

Author Response

Santiago, February 1, 2023

Guest Editor

Special Issue "Nutrition and Exercise Interventions on Skeletal Muscle Physiology, Injury and Recovery: From Mechanisms to Therapy"

Nutrients

We would like to thank you for letting us revise and resubmit the manuscript ID: nutrients-2186061. Title: The role of omega-3 polyunsaturated fatty acids and their lipid mediators on skeletal muscle regeneration: a narrative review”, sent for consideration for publication in Nutrients. The reviewers raised valuable comments and suggestions that helped us improve the manuscript.

We have responded to the points raised, explained our thoughts in this response letter, and revised the manuscript accordingly. We hope that you and the reviewers are pleased with our responses.

Answers and/or comments to the observations of the reviewer:

Reviewer #1

Point 1: This top-level review discusses the advantages of omega-3 fatty acids on muscle health.

Response 1: Thank you very much for your valuable and constructive comments.

Point 2: In the clinical part (paragraph 5), more emphasis should be put on the n-6/n-3 balance (as has been done in paragraph 3), as this explains (partially) the discrepancies between clinical studies.

Response 2: We appreciate this suggestion and now have added a sentence explaining that some of the discrepancies between clinical studies could result from an unbalance of the n-6/n-3 ratio.

Please see page 9; lines 389-391. It now reads: “Recent systematic reviews and meta-analyses have corroborated the efficacy of n-3 PUFAs in reducing DOMS and markers of muscle damage [103,104]; however, only  few studies have found significantly lower maximal muscle strength loss or recovery (i.e., maximal voluntary contraction; MVC), indicated as the best indirect marker of muscle damage, probably because of a poor control of the participants’ diet characteristics affecting the n-6/n-3 ratio and the use of broad range supplementation doses and times.

Point 3: While the n-6/n-3 balance is already quite well documented, I would also propose to add a short piece concerning n-7 and n-9 fatty acids.

Response 3: We thank you very much for this suggestion. We have added a new paragraph concerning the role of n-7 and n-9 fatty acids.

Please see pages 3-4; lines 125-136. It now reads: “Fatty acids (FAs) have diverse functions of physiological importance as they are major components of cellular membranes, precursors for the synthesis of bioactive lipids, and major sources of energy [30]. Saturated FAs contain no double bonds in their hydrocarbon chain, have a rigid structure, and are the most abundant FAs in the human diet with palmitic and stearic acid as the most popular. Conversely, unsaturated FAs have one or more double bonds in their hydrocarbon chain; those with one double bond are classified as monounsaturated fatty acids (MUFAs). The most common MUFAs are the omega-7 (n-7) and omega-9 (n-9), being palmitoleic acid and oleic acid as the most consumed in the diet, respectively. Both MUFAs have been reported to reduce the risk of heart disease and inflammation [31,32].”  

“FAs with two or more double bonds are classified as polyunsaturated fatty acids (PUFAs)…”

Sincerely yours,

Prof. Rodrigo Valenzuela. PhD.

Associate Professor

Department of Nutrition

Faculty of Medicine

University of Chile

Contact information: Rodrigo Valenzuela B. PhD, Nutrition Department, Faculty of Medicine, Universidad de Chile, Santiago, Chile, Independencia 1027, Casilla 70000, Santiago 7, Chile, Tel.: +56 2 29786014; Fax: +56 2 9786182; E-mail address: [email protected]  

Reviewer 2 Report

This is a well organized and lucid review of potential benefits and mechanisms of these benefits provided by dietary omega-3 supplementation on muscle damage and repair. The initial description of muscle damage and repair mechanisms, with emphasis on pro- and anti-inflammatory responses is appropriate and leads nicely into the discussion of potential effects of omega-3s. The last part of the review discussing current findings of effects of such supplementation in and animal and human models highlights the areas that require further research. This is a good comprehensive review

Author Response

Santiago, February 1, 2023

Guest Editor

Special Issue "Nutrition and Exercise Interventions on Skeletal Muscle Physiology, Injury and Recovery: From Mechanisms to Therapy"

Nutrients

We would like to thank you for letting us revise and resubmit the manuscript ID: nutrients-2186061. Title: The role of omega-3 polyunsaturated fatty acids and their lipid mediators on skeletal muscle regeneration: a narrative review”, sent for consideration for publication in Nutrients. The reviewers raised valuable comments and suggestions that helped us improve the manuscript.

We have responded to the points raised, explained our thoughts in this response letter, and revised the manuscript accordingly. We hope that you and the reviewers are pleased with our responses.

Answers and/or comments to the observations of the reviewer:

Reviewer 2

Point 1: This is a well organized and lucid review of potential benefits and mechanisms of these benefits provided by dietary omega-3 supplementation on muscle damage and repair. The initial description of muscle damage and repair mechanisms, with emphasis on pro- and anti-inflammatory responses is appropriate and leads nicely into the discussion of potential effects of omega-3s. The last part of the review discussing current findings of effects of such supplementation in and animal and human models highlights the areas that require further research. This is a good comprehensive review.

Response 1: Thank you very much for your comments. We are very happy that you found the review well organized and lucid.

Sincerely yours,

Prof. Rodrigo Valenzuela. PhD.

Associate Professor

Department of Nutrition

Faculty of Medicine

University of Chile

Contact information: Rodrigo Valenzuela B. PhD, Nutrition Department, Faculty of Medicine, Universidad de Chile, Santiago, Chile, Independencia 1027, Casilla 70000, Santiago 7, Chile, Tel.: +56 2 29786014; Fax: +56 2 9786182; E-mail address: [email protected]  
